# Genetic ablation of Cullin-RING E3 ubiquitin ligase 7 restrains pressure overload-induced myocardial fibrosis

**Melanie Anger**[1,2�], **Florian Scheufele**[1,2�], **Deepak Ramanujam**[1,2], **Kathleen Meyer**[1,2¤], **Hidehiro Nakajima**[3], **Loren J. Field**[3], **Stefan Engelhardt**[1,2], **Antonio Sarikas**[4]*

**1** Institute of Pharmacology and Toxicology, Technische Universität München, Munich, Germany, **2** German Center for Cardiovascular Research, Partner Site Munich Heart Alliance, Munich, Germany, **3** Wells Center for Pediatric Research and Krannert Institute of Cardiology, Indiana University School of Medicine, Indianapolis, Indiana, United States of America, **4** Institute of Pharmacology and Toxicology, Paracelsus Medical University, Salzburg, Austria

উ These authors contributed equally to this work.
¤ Current address: Cellular Plasticity and Disease Group, Institute for Research in Biomedicine, Barcelona, Spain
* antonio.sarikas@pmu.ac.at

**Data Availability Statement:** The data underlying the results presented in the study are available from Figshare at https://doi.org/10.6084/m9.figshare.13224341.v1.

## Abstract

Fibrosis is a pathognomonic feature of structural heart disease and counteracted by distinct cardioprotective mechanisms, e.g. activation of the phosphoinositide 3-kinase (PI3K) / AKT pro-survival pathway. The Cullin-RING E3 ubiquitin ligase 7 (CRL7) was identified as negative regulator of PI3K/AKT signalling in skeletal muscle, but its role in the heart remains to be elucidated. Here, we sought to determine whether CRL7 modulates to cardiac fibrosis following pressure overload and dissect its underlying mechanisms. For inactivation of CRL7, the Cullin 7 (Cul7) gene was deleted in cardiac myocytes (CM) by injection of adeno-associated virus subtype 9 (AAV9) vectors encoding codon improved Cre-recombinase (AAV9-CMV-iCre) in *Cul7$^{flox/flox}$* mice. In addition, Myosin Heavy Chain 6 (Myh6; alpha-MHC)-MerCreMer transgenic mice with tamoxifen-induced CM-specific expression of iCre were used as alternate model. After transverse aortic constriction (TAC), causing chronic pressure overload and fibrosis, AAV9-CMV-iCre induced *Cul7-/-* mice displayed a ~50% reduction of interstitial cardiac fibrosis when compared to *Cul7+/+* animals (6.7% vs. 3.4%, p<0.01). Similar results were obtained with *Cul7$^{flox/flox}$ Myh6-Mer-Cre-Mer$^{Tg(1/0)}$* mice which displayed a ~30% reduction of cardiac fibrosis after TAC when compared to *Cul7$^{+/+}$ Myh6-Mer-Cre-Mer$^{Tg(1/0)}$* controls after TAC surgery (12.4% vs. 8.7%, p<0.05). No hemodynamic alterations were observed. AKT$^{Ser473}$ phosphorylation was increased 3-fold (p<0.01) in *Cul7-/-* vs. control mice, together with a ~78% (p<0.001) reduction of TUNEL-positive apoptotic cells three weeks after TAC. In addition, CM-specific expression of a dominant-negative CUL7$^{1152stop}$ mutant resulted in a 16.3-fold decrease (p<0.001) of in situ end-labelling (ISEL) positive apoptotic cells. Collectively, our data demonstrate that CM-specific ablation of Cul7 restrains myocardial fibrosis and apoptosis upon pressure overload, and introduce CRL7 as a potential target for anti-fibrotic therapeutic strategies of the heart.

**Funding:** AS: Marie Curie International Reintegration Grant https://ec.europa.eu (#256584) The funders had no role in study design, data collection and analysis, decision to publish, or preparation of the manuscript.

**Competing interests:** The authors have declared that no competing interests exist.

# Introduction

The Ubiquitin-Proteasome System (UPS) is a selective protein degradation pathway that is involved in the pathogenesis of several cardiac disorders [1, 2]. Central to the UPS is the recognition of a substrate by an E3 ubiquitin ligase, a step pivotal for the ubiquitin-mediated degradation of substrate proteins by the 26S proteasome [3]. Cullin-RING complexes (CRLs) constitute the largest group of E3 ubiquitin ligases [4]. Cullin 7 (Cul7; formerly known as p193) is a component of the Cullin-RING E3 ubiquitin ligase 7 (CRL7), a multimeric enzyme composed of the RING finger protein ROC1, SKP1-FBXW8 substrate targeting subunit and CUL7 as scaffold protein [5–7]. In vitro, CRL7 was shown to target insulin receptor substrate 1 (IRS-1), a component of the insulin and insulin-like growth factor 1 (IGF1) signalling pathways, for ubiquitin-mediated degradation by the 26S proteasome [6, 8]. Mouse embryonic fibroblasts of Cul7-/- mice were found to accumulate IRS-1 and exhibit increased activation of IRS-1 downstream pathways phosphatidylinositol 3 (PI3)-kinase/AKT and MEK/ERK, respectively [6]. In vivo, heterozygosity of either Cul7 or Fbxw8 resulted in elevated PI3 kinase/AKT activation in skeletal muscle tissue upon insulin stimulation when compared to wild-type controls [9].

In the heart, Cul7 was initially identified as regulator of cardiac myocyte (CM) cell cycle activity in response to expression of DNA of tumorvirus oncoproteins [10–12]. Co-expression of dominant-interfering CUL7 and p53 mutants blocked E1A-induced apoptosis, resulting in robust CM proliferation [13]. In addition, CUL7 was shown to physically interact with Simian Virus 40 (SV40) Large T (LT) antigen oncoprotein and p53 [11, 12]. Transgenic expression of a dominant-interfering Cul7 mutant resulted in CM cell cycle re-entry at the infarct border zone four weeks after permanent coronary artery occlusion and was associated with an induction of DNA synthesis in the interventricular septa of infarcted hearts [14].

It was hypothesized that in CM, expression of dominant-interfering CUL7 mutants mimicked the effects of LT-antigen binding to CUL7, thereby altering its function [14]. We previously reported that binding of LT-antigen to CUL7 impairs CRL7 ubiquitin ligase function, resulting in the accumulation of the CRL7 substrate protein IRS-1 and upregulation of its downstream signalling pathways PI3K/AKT and MEK/ERK [15]. In the heart, PI3K/AKT plays an important cardioprotective role by regulating various cellular functions, e.g. CM survival and tissue remodelling [16]. Experimental activation of PI3K/AKT signalling resulted in reduced CM apoptosis during ischemia-reperfusion injury [17, 18] or pressure overload models [19]. In contrast, decreased PI3K/AKT signalling was linked to elevated apoptosis [20].

Based on these observations, we sought to investigate whether CRL7 contributes to the regulation of fibrosis and apoptosis in the heart.

# Materials and methods

## Cardiac myocyte-specific genetic ablation of Cul7

Transgenic mice carrying a homozygous LoxP-flanked Cul7 gene (*Cul7^flox/flox*) were kindly provided by J. DeCaprio and described previously [21]. For adeno-associated virus serotype 9 (AAV9)-mediated CM-specific genetic ablation of Cul7, AAV9 virions expressing a codon-improved Cre (iCre) recombinase [22] under the control of the CMV promoter (AAV9-CMV-iCre) were injected with a dose of 5 x 10^11 genome copies per mouse in the pericardial sack of *Cul7^flox/flox* mice at postnatal day 4/5 as described previously [23]. As alternate model, Myosin Heavy Chain 6 (Myh6; alpha-MHC)-MerCreMer transgenic mice with tamoxifen-induced CM-specific expression of iCre were used as described previously [24]. *Myh6-MerCreMer^Tg/0* mice were crossbred with *Cul7^flox/flox* mice to obtain *Cul7^flox/flox Myh6-MerCreMer^Tg/0*

and *Cul7⁺/⁺Myh6-MerCreMer^{Tg/0}* control littermates. At the age of 5–6 weeks, MerCreMer^{Tg/0} mice were either administered five daily injections of 40 mg/kg of tamoxifen (Sigma) or vehicle. After 2–8 weeks mice were euthanized for further analysis. Finally, transgenic mice with CM-specific expression of a dominant-negative CUL7 mutant under the control of the α-cardiac myosin heavy (MHC) promoter were utilized (MHC-1152stop, [14]).

## Generation of AAV vectors

Adeno-associated virus subtype 9 (AAV9) encoding iCre were prepared as described previously [23]. Briefly, infectious recombinant AAV vector particles were generated in HEK 293T cells by a cross-packaging approach whereby the vector genome was packaged into AAV9 capsid [25]. The rAAV titers, determined by measuring the copy number of viral genomes in pooled, dialyzed gradient fractions as described by Zentilin et al. [26], were in the range of $7 \times 10^{12}$ – $1 \times 10^{13}$ genome copies/ml. For AAV9-CMV-iCre production, HEK 293T cells were grown in triple flasks for 24 h (DMEM, 10% FCS). The AAV9-CMV-iCre backbone and helper (pDP9rs) plasmids were transfected into the HEK 293T cells using polyethylenimine (Sigma-Aldrich). The virus was purified after 72 hrs from benzonase-treated cell lysates over an iodixanol density gradient (Optiprep, Sigma-Aldrich). AAV titers were calculated by Real-time qPCR on vector genomes using FastStart Universal SYBR Green Master (Roche).

## Animal disease models

Transverse aortic constriction (TAC) was performed as described previously [23]. In sham surgery, only the chest was opened, but no ligation of the aorta was carried out.

Cardiac dimensions and function were analyzed by pulse-wave Doppler echocardiography before TAC/sham surgery and before the animals were euthanized. All animal studies were performed in accordance with the relevant guidelines and regulations and with approval of the responsible authorities (Regierung von Oberbayern, Munich, Germany; permit number 55.2-1-54-2532-160-13).

## Apoptosis assays

For terminal deoxynucleotidyltransferase-mediated dUTP nick-end labelling (TUNEL) analysis, paraffin sections were processed using the Situ Cell Death Detection Kit, TMR Red according to the procedures of the manufacturer for difficult tissue (Roche). DAPI was added for nuclear counterstaining. Images of whole-heart sections were acquired. Nuclei were automatically counted using an image analysis algorithm (MetaMorph). For in situ end labelling (ISEL) analysis, cryo-sections were processed using the KLENOW-FragEL DNA Fragmentation Detection Kit (Oncogene Research). Signal was developed with biotinylated nucleotides and horseradish peroxidase (HRP)-conjugated streptavidin, followed by incubation with diaminobenzidine.

## Cell culture and conditioned medium experiments

Murine CM and CF were isolated from 8-week-old adult male *Cul7+/+* and *Cul7-/-* animals as indicated and incubated in DMEM containing 10% FCS, 1% BrdU and 1% penicillin/streptomycin.

## Histochemical and immunohistological analyses

For analysis of collagen deposition, sections of left ventricular myocardium were fixed in 4% paraformaldehyde at room temperature (RT), embedded in paraffin and sectioned at 5 μm

intervals. Hematoxylin/eosin (H&E) and Sirius Red/Fast Green staining were performed as described previously [23].

## Immunoblot analysis

Protein extraction was performed in lysis buffer containing protease and phosphatase inhibitors (Roche) and immunoblots performed using standard procedures. Protein lysates were electrophoresed on 10% or 12% SDS-PAGE gels, transferred onto a PVDF membrane and blocked with BSA for two hours at RT.

## Quantification of mRNA in isolated cells or tissue

RNA was extracted from primary CM, CF or whole heart tissue using TriFast (PeqLab) and oligo-dT-primed cDNA synthesis was performed with Superscript II (Invitrogen), according to the manufacturer's instructions. Quantitative Real-time PCR amplification of *Cul7*, *Cyr61*, *Ctgf*, *Tgfb1*, *Mmp2*, *Mmp3*, *Mmp9*, *Pdgfa*, *Timp*, *Anp* and *Rpl32* was performed with primers listed in S1 Table using FastStart Universal SYBR Green Master (Roche). Data were normalized to *Rpl32* as indicated. Gene expression was analysed using hearts from 8-week-old male *Cul7-/-* mice or respectively 12-week-old *Cul7-/-* mice after TAC. Material from AAV9-CMV-dsRed injected littermates served as control.

## Wheat germ agglutinin staining

For determination of cardiomyocyte cross-sectional areas (CSA), 6 μm thick paraffin-embedded sections were prepared from murine hearts and stained with Alexa Fluor 647-conjugated wheat-germ agglutinin (WGA, Life Technologies, 1:100 dilution) for cell border determination and with SYTOX Green (Life Technologies, 1:1000 dilution) for nuclei detection. Images were taken from areas of transversely cut muscle fibers. The MetaMorph software (Molecular Devices) was programmed to recognize individual cells based on the WGA staining in an automated manner and proper thresholds were set for background and excessive fibrosis exclusion. MetaMorph's integrated morphometry analysis tool was used to calculate the average cell area of cardiomyocytes.

## Antibodies and reagents

For immunoblotting and histology, the following primary antibodies were used: anti-HSP90 (sc-13119, Santa-Cruz), anti-CUL7 (C1743, Sigma), anti-IRS-1 (LBC1863297, Millipore), anti-AKT (9272S, Cell Signalling), anti-P-AKT $^{S473}$ (9271S, Cell Signalling), anti-ERK1/2 (4696S, Cell Signalling), anti-P-ERK1/2 $^{T202/Y204}$ (9101S, Cell Signalling). Secondary horseradish peroxidase or fluorophore-conjugated antibodies were from Life technologies (Carlsbad, CA). Penicillin, streptomycin, fetal bovine serum (FCS) and other reagents were obtained from Invitrogen Life Technologies (Carlsbad, CA) or Sigma-Aldrich (Louis, MO).

## Statistical analysis

Data are shown as mean ± SEM. Statistical analysis was performed with Prism (GraphPad software, version 6). Differences between two means were assessed by unpaired t-test. Differences among multiple means were assessed by 1-way or 2-way ANOVA followed by the Bonferroni correction as indicated. A *P*-value of $< 0.05$ was considered significant.

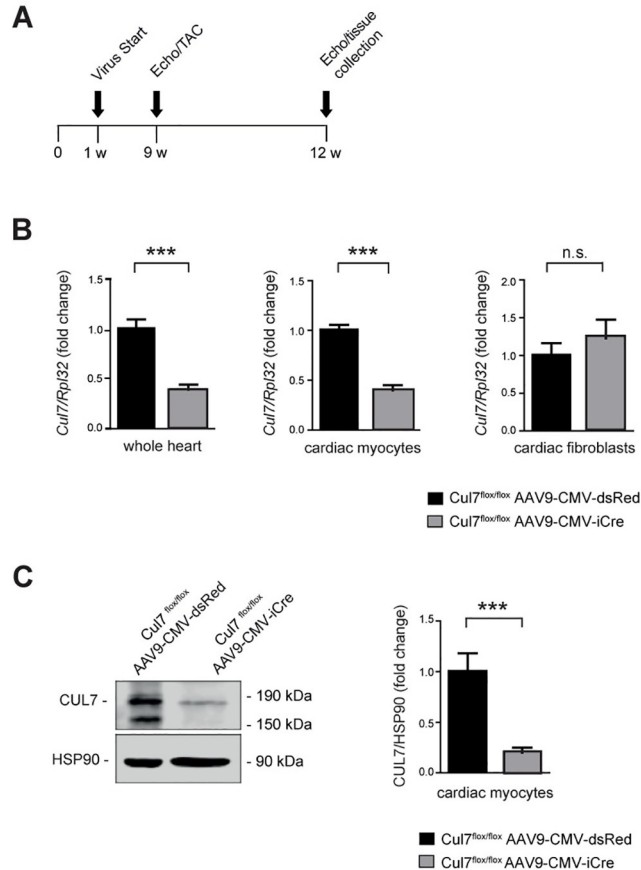

**Fig 1. Generation and validation of cardiomyocyte-specific *Cul7*-/- mice.** Experimental strategy and timeline (**A**). On day 4 to 5 after birth mice were transduced pericardially with AAV9-CMV-iCre or AAV9-CMV-dsRed serving as control group. At week 9 mice underwent sham or TAC surgery und were euthanized for further analysis at week 12. (**B**) Realtime-PCR of *Cul7* mRNA in whole heart lysates (WH, left panel, n = 7–9 mice/group), isolated cardiac myocytes (CM, middle panel, n = 7 mice/group) and isolated cardiac fibroblasts (CF, right panel, n = 7–9 mice/group). (**C**) Immunoblot analysis for CUL7 protein in isolated CM (left panel) and quantification thereof (right panel). n = 3–7 mice/group. Data are shown as fold change normalized to controls and expressed as mean ± SEM. ***p<0.001 (unpaired t-test). WH: whole heart, CM: cardiac myocytes, CF: cardiac fibroblasts.

## Results

### Generation of cardiac myocyte-specific Cul7 knockout mice

To investigate the role of CRL7 in the heart, two different Cre/LoxP-based CM-specific genetic knockout animal models of Cul7, the core component of CRL7 [5], were generated. Adeno-associated virus serotype 9 (AAV9) mediated gene transfer of codon-enhanced Cre-recombinase (iCre) was chosen for its strong heart tropism [27, 28]. Male *Cul7^{flox/flox}* mice received AAV9-CMV-iCre or AAV9-CMV-dsRed ($5 \times 10^{11}$ genome copies per mouse) by pericardial injection on postnatal day 4–5 (Fig 1A). After 8 weeks, knockdown efficacy was confirmed by quantitative Realtime-PCR analysis. *Cul7* mRNA level were reduced by 60.8 ± 0.1% (p<0.001) in whole heart extracts (WH) of *Cul7^{flox/flox}AAV9-CMV-iCre* mice when compared to AAV9-CMV-dsRed control virus treated mice (Fig 1B). To confirm cardiac myocyte-specific ablation of Cul7 upon AAV-iCre gene transfer, hearts were subjected to collagenase digestion with subsequent separation of myocytes and non-myocytes 8 weeks after injection of AAV-iCre.

Importantly, *Cul7* mRNA was reduced by 60.1 ± 0.1% (p<0.001) in CM but remained unaltered in cardiac fibroblasts (CF) of *Cul7^{flox/flox}AAV9-CMV-iCre* mice, when compared to AAV9-CMV-dsRed control virus treated mice (Fig 1B). Finally, immunoblot analysis revealed markedly reduced CUL7 protein concentrations (-79.9 ± 0.1%; p<0.001) in purified CM of *Cul7^{flox/flox}AAV9-CMV-iCre* mice when compared to Cul7^{flox/flox}AAV9-CMV-dsRed controls (Fig 1C).

As alternative CM-specific Cul7 knockout model we used Myosin Heavy Chain 6 (Myh6;α-MHC)-MerCreMer transgenic mice with tamoxifen-induced CM-specific expression of Cre, initially described by Molketin and colleagues [24]. *Myh6-MerCreMer^{Tg/0}* mice were crossbred with *Cul7^{flox/flox}* mice to obtain *Cul7^{flox/flox} Myh6-MerCreMer^{Tg/0}* and *Cul7^{+/+}Myh6-MerCreMer^{Tg/0}* control littermates. At the age of 6 weeks, *MerCreMer^{Tg/0}* mice were either administered injections of 40 mg/kg of tamoxifen or vehicle for five consecutive days and animals euthanized for further analysis at week 12 (S1A Fig). Recombination events were confirmed by PCR (S1B Fig) and immunoblot analysis confirmed a marked loss of CUL7 protein in whole heart extracts of *Cul7^{flox/flox}Myh6-MerCreMer^{Tg(1/0)}* mice when compared to *Cul7^{+/+}Myh6-MerCreMer^{Tg(1/0)}* mice (-71.3 ± 0.2%, p<0.01) (S1C Fig). In addition, CUL7 protein depletion was cell type-specific for CM, but not CF, of *Cul7^{flox/flox}Myh6-MerCreMer^{Tg(1/0)}* mice (S1D and S1E Fig).

Functional phenotypisation of AAV9-CMV-iCre and *Cul7^{flox/flox}Myh6-MerCreMer^{Tg/0}* mice at basal conditions by pulse-wave Doppler echocardiography revealed no statistically significant differences when compared to control littermates (S2 and S3 Figs). However, a minor but statically significant increase of heart weight to body weight and heart weight to tibia length ratio concomitant with a decrease in left ventricular volume was observed with AAV9-CMV-iCre treated mice (S2 Fig).

## Loss of Cul7 attenuates cardiac fibrosis upon pressure overload

To determine the impact of genetic ablation of Cul7, thereby disrupting the CRL7 ubiquitin ligase complex, on cardiac fibrogenesis, we performed transverse aortic constriction (TAC), a well-established animal disease model for pressure overload-induced cardiac hypertrophy and fibrosis [29]. 9-week-old *Cul7^{flox/flox}AAV9-CMV-iCre* or *Cul7^{flox/flox}AAV9-CMV-dsRed* mice were subjected to TAC or control (sham) surgery. At week 12, animals were euthanized and heart samples were collected. A robust induction of myocardial hypertrophy upon TAC was confirmed by realtime-PCR of atrial natriuretic peptide (ANP) and quantification of CM cross sectional area (S11–S13 Figs). Strikingly, hearts of *Cul7^{flox/flox}AAV9-CMV-iCre* mice displayed a marked ~50% reduction in fibrosis (6.7 ± 1.1% vs. 3.4 ± 0.9%, p<0.01) when compared to *Cul7^{flox/flox}AAV9-CMV-dsRed* control mice 3 weeks after TAC surgery (Fig 2A and 2B). Cardiac fibrosis, as evidenced by Sirius Red / Fast Green staining, was attenuated to a similar degree in areas of interstitial and perivascular fibrosis in TAC-hearts of *Cul7^{flox/flox}AAV9-CMV-iCre* mice when compared to control littermates (Fig 2A). In corroboration of these findings, *Cul7^{flox/flox}Myh6-Mer-Cre-Mer^{Tg(1/0)}* mice displayed a ~30% reduction of interstitial and perivascular fibrosis after TAC when compared to *Cul7^{+/+};Myh6-Mer-Cre-Mer^{Tg(1/0)}* controls 4 weeks after TAC surgery (12.4% vs. 8.7%, p<0.05) (S4 Fig).

Of note, functional phenotypisation of AAV9-CMV-iCre (S5 and S6 Figs) and *Cul7^{flox/flox}Myh6-MerCreMer^{Tg/0}* (S7 and S8 Figs) mice after TAC by pulse-wave Doppler echocardiography revealed no statistically significant differences when compared to control littermates.

Collectively, phenotypisation of two different genetic CRL7 loss-of-function animal models revealed that ablation of Cul7 results in a marked attenuation of cardiac fibrosis after TAC, indicating that CRL7 acts as regulator of fibrosis in the pressure-overloaded heart.

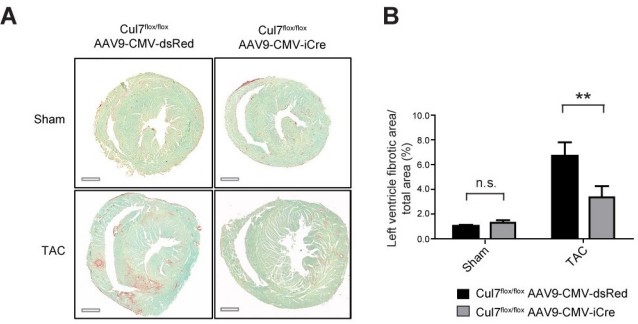

**Fig 2. Genetic ablation of *Cul7* restrains cardiac fibrosis under conditions of pressure overload.** 9-week-old *Cul7^{flox/flox};AAV9-CMV-iCre* or *Cul7^{flox/flox};AAV9-CMV-dsRed* mice were subjected to TAC or control (sham) surgery. At week 12, animals were euthanized and heart samples collected. **(A)** Representative images of transverse heart sections stained for fibrosis (Sirius red/fast green) after sham (upper panel) or TAC (lower panel) surgery. **(B)**. Quantification of results. Scale bar = 1 mm. Data are shown as mean ± SEM; n = 5–6 mice/group. **p<0.01 (two-way ANOVA).

## Enhanced activation of PI3K/AKT signalling in cardiac myocytes depleted of Cul7

The CRL7 ubiquitin ligase complex was previously identified as regulator of the PI3K/AKT and MEK/ERK signalling pathways through ubiquitin-mediated degradation of IRS-1 [6]. Heterozygosity of either Cul7 or Fbxw8, both key components of CRL7, resulted in over-activation of PI3K/AKT signalling in skeletal muscle upon insulin stimulation [9]. To further delineate underlying molecular mechanisms of the anti-fibrotic phenotype observed in CM-specific Cul7 knockout mice, we investigated the abundance and activation of CRL7-regulated proteolytic target proteins and signalling pathways, respectively, in CM lysates by immunoblot analysis. Strikingly, CM lysates of *Cul7^{flox/flox}AAV9-CMV-iCre* mice showed an ~3-fold (p<0.01) increase of AKT^{Ser473} phosphorylation when compared to *Cul7^{flox/flox} AAV9-CMV-dsRed* control mice (Fig 3A). IRS-1 protein level was increased ~2-fold, albeit not statistically significant due to high variation between mice (p = 0.1705) (Fig 3B). Phosphorylation of ERK1/2 at Thr202/Tyr204 was moderately increased (~1.5-fold increase, p = 0.2555) (Fig 3C). Similar results were obtained with CM lysates of *Cul7^{flox/flox}Myh6-Mer-Cre-Mer^{Tg(1/0)}* mice (S9 Fig). Collectively, these data confirm our previous findings of CRL7 as regulator of PI3K/AKT activation and provides first evidence for a physiological role of CRL7 in regulating PI3K/AKT signalling in the heart.

## Loss of Cul7 results in reduced apoptosis of cardiac myocytes

Loss of CM, e.g. by apoptosis, is a hallmark of myocardial remodelling at conditions of sustained overload, leading to replacement fibrosis [30]. As PI3K/AKT signalling is a major pro-survival pathway to protect CM in states of cellular stress [16], we hypothesized that attenuated fibrosis in the CRL7 loss-of-function models are due to reduced CM apoptosis. To test this hypothesis, hearts of *Cul7^{flox/flox}AAV9-CMV-iCre* and *Cul7^{flox/flox} AAV9-CMV-dsRed* were subjected to TUNEL-staining three weeks after TAC surgery. Remarkably, myocardial tissue of *Cul7^{flox/flox}AAV9-CMV-iCre* mice showed a ~4.5-fold reduction in TUNEL-positive (apoptotic) cells when compared to controls (0.6 ± 0.3% vs. 2.7 ± 0.3%; p<0.001) (Fig 4). To further substantiate these findings, we analysed heart samples from Isoprenaline (ISO)-treated transgenic mice with CM-specific expression of the dominant-negative CUL7^{1152stop} mutant [14]. For apoptosis quantification, heart sections of MHC-1152stop mice were subjected to in situ

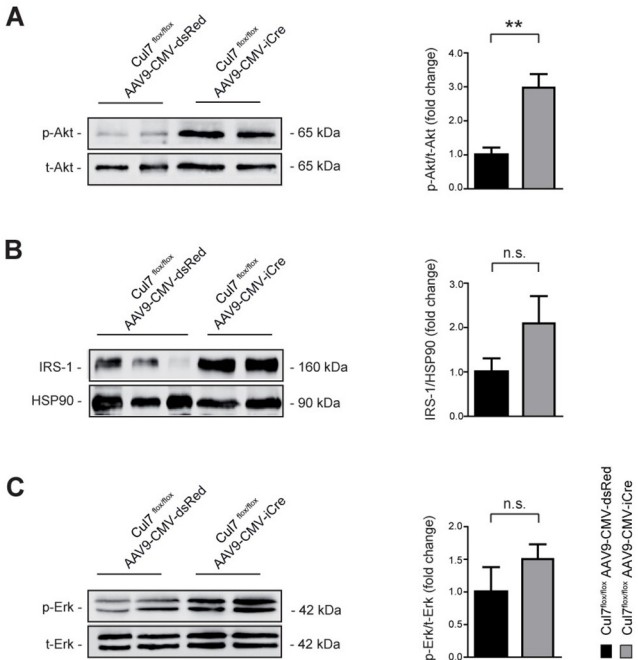

**Fig 3. Increased PI3K/AKT signalling in *Cul7*-depleted cardiac myocytes.** Lysates of CM isolated from *Cul7^flox/flox^AAV9-CMV-iCre* mice at 8 weeks of age were subjected to SDS-PAGE and Western blot analysis using antibodies directed against AKT $^{PS473}$ **(A)**, IRS-1 **(B)**, ERK $^{PT202/pY204}$ **(C)** and HSP90. Data are shown as fold change normalized to CM of age- and sex-matched *Cul7^flox/flox^AAV9-CMV-dsRed* mice and expressed as mean ± SEM. n = 4-7/group. ***p<0.01 (unpaired t-test).

end-labelling (ISEL) analysis. In line with our above findings, we observed 16.3-fold decrease (0.004% vs. 0.065%, p <0.001) of ISEL-positive apoptotic cells in hearts of MHC-1152stop when compared to control mice (MHC-nLAC/-) mice (S2 Table). Of note, ISO treatment did not result in increased S-phase activity in transgenic mice expressing a dominant-interfering Cul7 mutant in CM (S2 Table). Collectively, the above data support a role for CRL7 in pro-

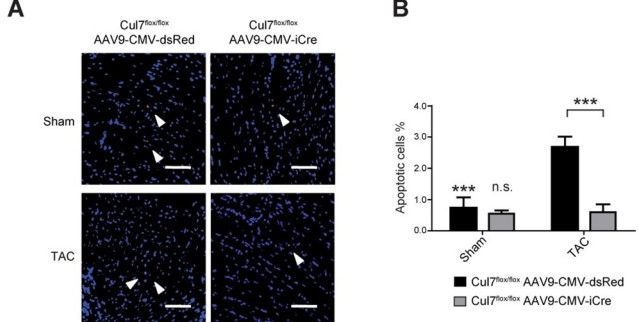

**Fig 4. *Cul7*-/- mice are less prone to apoptosis induction in the heart upon pressure overload.** 9-week-old *Cul7^flox/flox^AAV9-CMV-iCre* or *Cul7^flox/flox^AAV9-CMV-dsRed* mice were subjected to TAC or control (sham) surgery. At week 12, animals were euthanized and heart samples collected. Apoptosis was analysed in paraffin sections of mouse hearts by TUNEL staining. **(A)** Representative pictures of TUNEL staining after sham (upper panel) or TAC-surgery (lower panel), scale bar = 100 μm. **(B)** Quantification of results. Data are shown as mean ± SEM. ***p<0.001 (two-way ANOVA).

survival signalling of CMs and suggests that inhibition of CRL7-mediated PI3K/AKT signalling protects against pressure overload-induced apoptosis in the heart.

## Reduced myocardial expression of Tgfb1 upon genetic ablation of Cul7

To further delineated mechanisms of CRL7-mediated fibrotic remodelling of the heart, we studied the expression of a subset of genes encoding pro- and anti-fibrotic factors in CM upon genetic ablation of Cul7. Hearts of 12-week-old *Cul7flox/flox; AAV9-CMV-iCre* were harvested three weeks after TAC surgery and snap-frozen. Relative gene expression of transforming growth factor beta 1 (*Tgfb1)*, cysteine rich protein 61 (*Cyr61*), connective tissue growth factor (*Ctgf*), matrix metallopeptidase 2 (*Mmp2*), matrix metallopeptidase 3 (*Mmp3*), matrix metallopeptidase 9 (*Mmp9*), platelet derived growth factor alpha (*Pdgfa*) and tissue inhibitor of metalloproteinase 1 (*Timp1*) was measured by quantitative RT-PCR. Surprisingly, of all candidate genes analysed, only *Tgfb1* transcripts were markedly changed (- 68 ± 0.2%, p<0.05) in TAC hearts when compared to *Cul7flox/flox*AAV9-CMV-dsRed control animals (S10 Fig). These data suggest that altered *Tgfb1* gene expression may contribute to the anti-fibrotic phenotype observed in Cul7 knockout hearts after TAC.

## Discussion

This study characterized the role of CRL7 in cardiac remodelling and myocardial fibrosis. By using different genetic knockout animal models, we demonstrate that CM-specific ablation of Cul7 results in markedly reduced interstitial fibrosis upon pressure overload. Loss of Cul7 led to increased activation of the PI3K/AKT signalling pathway, that was accompanied with a significant reduction of myocardial cell apoptosis after TAC. Finally, we showed that targeted ablation of Cul7 in CM significantly downregulated mRNA level of fibrosis-modifying gene Tgfb1 under conditions of increased afterload. Collectively, our data indicate that Cul7 acts as a pro-fibrotic factor in the heart during conditions of increased myocardial stress, e.g. pressure overload.

Previous studies revealed a critical role for Cul7 in the heart in the context of cell cycle control [14]. Transgenic expression of a dominant-interfering Cul7 mutant (Cul7[MHC-1152] stop mice) resulted in a marked induction of CM cell cycle activity at the infarct border zone 4 weeks after permanent coronary artery occlusion and was associated with an induction of cardiomyocyte DNA synthesis in the interventricular septa of infarcted hearts [14]. Although cell cycle activity was not measured in the cardiac-restricted Cul7 null models employed in the present study, it is important to note that ISO treatment did not result in an increase in cardiomyocyte S-phase activity in transgenic mice expressing dominant-interfering Cul7 vs. WT mice. Thus, the marked reduction in cardiomyocyte apoptosis in these ISO-treated transgenic mice must occur independently of cardiomyocyte S-phase activity. Interestingly, expression of the MHC-1152stop mutant was also associated with a decreased infarct scar expansion in the septa of hearts, suggesting that CRL7 inactivation might partially counteract the adverse ventricular remodelling that occurs after infarction [14]. In a follow-up study, Hassink et al. corroborated these findings and showed that 4 weeks after myocardial infarction, ventricular isovolumic relaxation time constant (Tau) was decreased by 19% (p<0.05) and the slope of the dP/d*t*max-EDV relationship was increased 99% (p<0.05) in infarcted MHC-1152stop mice when compared to infarcted non-transgenic littermates [31], suggesting a significant improvement in cardiac function. Our data further expands these findings by providing evidence for a pro-fibrotic role of Cul7 in the context of pressure overload-induced fibrosis. Interestingly, a recent study reported a pro-fibrotic role of Cul7 in patients with hepatocellular carcinoma and amplification of the 6p21.1 gene locus [32]. In these patients, elevated CUL7 protein

concentrations were strongly correlated with advanced fibrosis in peritumoral liver areas. Collectively, these data suggest that CRL7 regulates general pro-fibrotic mechanisms that are effective in different organ systems.

What is the molecular mechanism of Cul7-mediated fibrogenesis? Pathological remodelling of the myocardium is typically characterized by loss of CM due to apoptosis, necrosis or phagocytosis [30]. We observed a marked reduction of apoptosis in Cul7-deficient CM, associated with a 3-fold over-activation of the PI3K/AKT signalling pathway. These observations are in concordance with previous reports using heterozygous conventional gene knockout models of Cul7 and Fbxw8 (the substrate targeting unit of the Cul7 ubiquitin ligase) [9]. In skeletal muscle tissue of *Cul7*$^{+/-}$ and *Fbxw8*$^{+/-}$ mice, AKT signalling was ~2-fold increased under basal conditions and up to ~6-fold under insulin stimulation when compared to control animals [9].

There is abundant evidence that activation of PI3K/AKT signalling has cardioprotective effects by limiting apoptotic cell death in the myocardium. Matsui et al. showed that adenoviral gene transfer of activated AKT protects CM from apoptosis in response to hypoxia in vitro [33]. Moreover, adenovirus-mediated AKT gene transfer in the heart diminishes CM apoptosis and limits infarct size following ischemia/reperfusion injury [34] and ameliorates doxorubicin-induced contractile dysfunction [35]. It is thus tempting to speculate that CRL7-mediated regulation of PI3K/AKT signalling contributes to CM survival, thereby preventing replacement fibrosis. However, we cannot rule out that CRL7 also affects other pro-fibrotic pathways independent of cell death.

It remains an open question whether CUL7 modulates apoptosis in a non-proteolytic manner, e.g. independent of CRL7's role in ubiquitin-mediated degradation of IRS-1. In a previous report, Tsai et al. identified a putative BH3 domain in the C-terminus of CUL7 [36] and postulated that CUL7 belongs to the BH3-only family of pro-apoptotic proteins. It was shown that forced expression of CUL7 in NIH-3T3 cells promoted apoptosis in a manner that was dependent on the integrity of the BH3 domain [36]. However, it should be noted that Kim et al. identified CUL7 in a functional screen for inhibitors of Myc-induced apoptosis, showing that expression of CUL7 prevented both c-Myc and N-Myc mediated apoptosis and promoted the transformation of neuroblastoma SHEP cells in a p53-dependent manner [37]. Further studies are therefore needed to dissect the different roles of CUL7 as a regulator of apoptosis in various cell types and tissues.

We observed a marked down-regulation of Tgfb1 gene transcripts in CM deleted of Cul7. Substantial evidence suggests that TGF-ß triggers fibrosis and apoptosis in different cell types [38]. Interestingly, expression of activated PI3K, AKT or IRS-1 was shown to antagonize TGF-ß-mediated apoptosis [39–41]. It was suggested that this pro-survival effect is dependent on PI3K/AKT-mediated suppression of Smad3, a central component of the TGF-ß signalling pathway [42]. It is tempting to speculate that over-activation of PI3K/AKT, as shown for *Cul7*-/- hearts, contributes to diminished Tgfb1 mRNA levels, thereby ameliorating fibrosis after TAC. In addition, PI3K/AKT signalling pathways regulate key events in the inflammatory response to damage [43]. It is therefore tempting to speculate that the observed effect of CRL7 ablation on cardiac fibrosis is partly mediated by PI3K/AKT-regulated inflammation and its downstream effects on fibrogenesis. Clearly, more studies are needed to further dissect the molecular mechanisms by which CRL7 restrains cardiac fibrosis. Whether pharmacological targeting of CRL7 is a useful strategy to mitigate excess fibrosis in the heart awaits future investigation.

It is generally believed that myocardial fibrosis promotes ventricular dysfunction, e.g. by increased diastolic stiffness and impaired relaxation (reviewed in [30]). However, an unexpected result of our study was the lack of functional cardiac improvement in Cul7-deficient

mice despite a ~50% reduction of interstitial myocardial fibrosis under conditions of pressure overload. A similar observation was made by Yang and colleagues who reported that the anti-fibrotic peptide Ac-SDKP markedly attenuated cardiac fibrosis without improving EF and E/A ratios [44]. It is tempting to speculate that CRL7-mediated modulation of fibrosis early in cardiac injury may affect cardiac repair and remodeling in an adverse manner, e.g. by suppression of protective collagen-mediated myocyte-to-myocyte interaction or alterations of the myocardial microenvironment. In addition, loss of regulated AKT activity upon genetic ablation of CRL7 may have affected cardiac function, as balanced AKT signaling is essential for contractility and the response to pathological stress [16]. Proof-of-concept studies have shown that CRLs are druggable targets. Wu et al. recently reported the identification of the small molecule drug suramin as inhibitor of CRLs that contain cullin 2, 3, and 4A [45]. Whether targeting CRL7 by small molecules will provide a novel strategy to impede excess fibrosis in the heart awaits future investigation. Further work is needed to dissect the molecular mechanisms of CRL7-mediated regulation of pressure overload-induced myocardial fibrosis and the potential of the CRL7 ubiquitin ligase as a pharmacological target for the treatment of cardiac fibrosis.

## Supporting information

**S1 Fig. Tamoxifen-induced genetic ablation of Cul7 in murine cardiac myocytes. (A)** Experimental strategy and timeline. **(B)** For checking recombination efficacy, a recombination PCR was performed followed by gel electrophoresis. Events of Cul7 allele recombination were only detectable in $Cul7^{flox/flox}$ $Myh6\text{-}MerCreMer^{Tg(1/0)}$ mice (lanes 10–13), when compared to $Cul7^{+/+}$ $Myh6\text{-}MerCreMer^{Tg(1/0)}$ (lanes 6–9) and $Cul7^{flox/flox}$ $Myh6\text{-}MerCreMer^{Tg(0/0)}$ (lanes 2–5). **(C)** Significant reduction of the Cullin7 protein abundance in whole hearts samples under basal conditions. n = 4 mice/group. **(D)** Lysates of isolated cardiac myocytes (CM) were subjected to immunoblot analyses for CUL7 protein. HSP90 served as internal control. Representative blot depicting CM-specific CUL7 knockdown and quantification. n = 4 mice/group. **(E)** Representative immunoblot of non-CM for CUL7 abundance (left) and quantification thereof (right). n = 4 mice/group. Data are shown as mean ± SEM. Unpaired t-test. *p<0.05, **p<0.01.
(TIF)

**S2 Fig. Cardiac phenotypisation and echocardiographic assessment of heart function of Cul7-/- mice under basal conditions injected with viral vectors.** Examination of 8-week-old mice compared to respective control animals injected with AAV9-CMV-dsRed under basal conditions. **(A)** Screening for cardiac hypertrophy by assessment of heart weight/ body weight-, heart weight/ tibia length-, and lung weight/ tibia length-ratio. n = 5–8 mice/group; *p<0.05, **p<0.01 (unpaired t-test). **(B)** Echocardiographic assessment of 8-week-old mice under basal conditions concerning measurement of fractional shortening, ejection fraction, left ventricular systolic and diastolic volume as well as left ventricular inner diameter in diastole and systole (LVID d/s). n = 8–11 mice/group; *p<0.05 (unpaired t-test).
(TIF)

**S3 Fig. Morphometric and functional phenotyping after loss of Cul7 under basal conditions. (A)** Morphometric assessment of heart weight/tibia length (HW/TL), ventricular weight/tibia length (VW/TL) and lung weight/tibia length (LW/TL) under basal conditions. **(B)** Functional assessment of fractional shortening, ejection fraction, left ventricular inner diameter in systole and diastole (LVID$_s$ resp. LVID$_d$) under basal conditions. n = 7–9 mice/group. Data are shown as mean ± SEM. *p<0.05, **p<0.01 (unpaired t-test).
(TIF)

**S4 Fig. Loss of Cul7 results in less cardiac fibrosis under conditions of increased afterload.** **(A)** Representative myocardial tissue sections after staining with Sirius Red (for collagen) and Fast Green counterstaining after transverse aortic constriction (TAC; lower panel). Mice were sacrificed after 4 weeks of increased afterload at an age of 12 weeks. Scale bar: 1 mm. **(B)** Quantification of myocardial fibrosis of sham vs. TAC mice. n = 7–9 mice/group. Data are shown as mean ± SEM. Two-way ANOVA. *p<0.05.
(TIF)

**S5 Fig. Evaluation of Cul7 gene expression and cardiac morphology after transverse aortic constriction and sham surgery in the AAV-approach.** **(A)** Quantitative qPCR analysis for Cul7 gene expression in whole heart samples 3 weeks after TAC or sham surgery. n = 4–5 mice/group; *p<0.05, **p<0.01 (student's t-test). Heart weight/ body weight- **(B)**, heart weight/ tibia length- **(C)** and lung weight/ tibia length-ratios **(D)**. n = 5–6 mice/group; Sham vs. TAC **(B-C)**: highly statistically significant (two-way ANOVA).
(TIF)

**S6 Fig. Cardiac morphometric dimensions and function 3 weeks after TAC in mice ablated from Cul7 via AAV9-CMV-iCre.** Heart function of mice injected with AVV9 was analysed by pulse-wave Doppler echocardiography upon sham as well as TAC surgery after 3 weeks of increment of afterload. Lane 1–2 sham cohort, lane 3–4 TAC cohort. **(A)** Functional parameters FS and EF after operation. Sham vs. TAC: highly statistically significant (two-way ANOVA). **(B)** Left ventricular inner diameter in systole as well as diastole after operation of the sham and the respective TAC cohort. **(C)** Systolic and diastolic left ventricular volume. n = 4–6 mice/group; Sham vs. TAC: not statistically significant (two-way ANOVA).
(TIF)

**S7 Fig. Morphometric phenotyping after loss of Cul7 under conditions of increased afterload.** Morphometric assessment of **(A)** heart weight/tibia length (HW/TL), **(B)** ventricular weight/tibia length (VW/TL), **(C)** atrial weight/tibia length (AW/TL), **(D)** lung weight/tibia length (LW/TL) and **(E)** body weight under basal conditions (sham) and conditions of increased afterload (TAC). n = 7–9 mice/group. Data are shown as mean ± SEM. *p<0.05, **p<0.01 (two-way ANOVA).
(TIF)

**S8 Fig. Functional phenotyping after loss of Cul7 under conditions of increased afterload.** Lane 1–2 sham cohort, lane 3–4 TAC cohort. Functional assessment of **(A)** fractional shortening, **(B)** ejection fraction, **(C, D)** left ventricular inner diameter in systole and diastole (LVID$_s$ resp. LVID$_d$) under basal conditions. n = 7–9 mice/group. Data are shown as mean ± SEM. *p<0.05, **p<0.01 (two-way ANOVA).
(TIF)

**S9 Fig. Increased AKT phosphorylation upon cardiac myocyte-specific Cul7 depletion.** 8-week old mice were sacrificed 2 weeks after intraperitoneal injection of Tamoxifen. **(A)** Representative immunoblot showing IRS1 protein levels, HSP90 served as internal control (left); quantification (right). n = 7 mice/group. **(B)** Immunoblot analysis of p-AKT and total AKT. n = 7 mice/group. **(C)** Immunoblot analysis of p-ERK1/2 and total ERK1/2. n = 7 mice/group. All data are shown as fold change normalized to *Cul7*[+/+] *Myh6-MerCreMer*[Tg(1/0)] controls and expressed as mean ± SEM. Unpaired t-test. *p<0.05.
(TIF)

**S10 Fig. Candidate gene expression hearts of *Cul7*[flox/flox] *AAV9-CMV-iCre* and control mice upon conditions of increased afterload.** Hearts of 12-week-old *Cul7*[flox/flox] *AAV9-CMV-*

*iCre* were harvested 3 weeks after TAC surgery, snap-frozen and analysed by quantitative RT-PCR. Relative gene expression was determined by normalization against *Rpl32* as described in Materials and Methods and expressed as fold changes relative to control samples of *Cul7$^{flox/flox}$ AAV9-CMV-dsRed* mice **(A-H)**. Values are means ± SEM, n = 3–6 mice/group. *p<0.05 (unpaired t-test).
(TIF)

**S11 Fig. Gene expression of atrial natriuretic peptide (ANP) in sham vs. TAC animals.** Quantification of ANP mRNA concentrations by realtime-PCR four (A) and six (B) weeks after transverse aortic constriction (TAC). Sham-treated littermates served as control. Rpl32 mRNA level served as internal controls. t- test; *: p < 0.05.
(TIF)

**S12 Fig. Cardiac myocytes cross sectional area of sham- or TAC-treated Cul7$^{flox/flox}$AAV9-CMV-iCre or Cul7$^{flox/flox}$AAV9-CMV-dsRed mice. (A)** Wheat germ agglutinin staining of representative myocardial sections of sham-operated animals (upper panel) and mice subjected to transverse aortic constriction (lower panel). **(B)** Quantification of cross sectional area. n = 5–6 mice/group; sham vs. TAC: ** P < 0.01, *** P < 0.001 (two-way ANOVA, Bonferroni post-test); scale bar = 100 μm.
(TIF)

**S13 Fig. Cardiac myocytes cross sectional area of sham- or TAC-treated Myh6-Mer-Cre-Mer$^{Tg(1/0)}$ or Cul7$^{flox/flox}$ Myh6-Mer-Cre-Mer$^{Tg(1/0)}$ mice. (A)** Wheat germ agglutinin staining of representative myocardial sections of sham-operated animals (upper panel) and mice subjected to transverse aortic constriction (lower panel). **(B)** Quantification of cross sectional area. n = 5–6 mice/group; sham vs. TAC: * P < 0.05, ** P < 0.01, *** P < 0.001 (two-way ANOVA, Bonferroni post-test); scale bar = 100 μm.
(TIF)

**S1 Table. Primer sequences.**
(DOCX)

**S2 Table. ISEL and cell cycle analysis.**
(DOCX)

## Acknowledgments

We thank James DeCaprio (Dana-Farber Cancer Institute, Boston, USA) for generously providing Cul7$^{flox/flox}$ mice and Jeffrey Molkentin (Cincinnati Children's Hospital Medical Center, Cincinnati, USA) for generously providing Myh6-MerCreMer$^{Tg/0}$ mice. We thank Zhen-Qiang Pan (Mount Sinai School of Medicine, New York, USA) for critical discussions. We thank Sabine Brummer for performing cardiac histology, Kornelija Sakac for echocardiographic analysis and TAC surgeries, Urszula Kremser for primary cell isolations.

## Author Contributions

**Conceptualization:** Melanie Anger, Florian Scheufele, Loren J. Field, Stefan Engelhardt, Antonio Sarikas.

**Data curation:** Melanie Anger, Florian Scheufele, Deepak Ramanujam, Kathleen Meyer, Hidehiro Nakajima, Loren J. Field, Antonio Sarikas.

**Formal analysis:** Melanie Anger, Florian Scheufele, Deepak Ramanujam, Kathleen Meyer, Hidehiro Nakajima, Antonio Sarikas.

**Funding acquisition:** Antonio Sarikas.

**Investigation:** Melanie Anger, Florian Scheufele, Kathleen Meyer, Hidehiro Nakajima, Loren J. Field, Antonio Sarikas.

**Methodology:** Melanie Anger, Florian Scheufele, Deepak Ramanujam, Kathleen Meyer, Hidehiro Nakajima, Loren J. Field, Antonio Sarikas.

**Project administration:** Melanie Anger, Florian Scheufele, Kathleen Meyer, Antonio Sarikas.

**Resources:** Antonio Sarikas.

**Software:** Melanie Anger, Florian Scheufele, Kathleen Meyer, Antonio Sarikas.

**Supervision:** Loren J. Field, Antonio Sarikas.

**Validation:** Kathleen Meyer, Antonio Sarikas.

**Visualization:** Deepak Ramanujam, Kathleen Meyer, Hidehiro Nakajima, Antonio Sarikas.

**Writing – original draft:** Antonio Sarikas.

**Writing – review & editing:** Melanie Anger, Florian Scheufele, Kathleen Meyer, Hidehiro Nakajima, Loren J. Field, Stefan Engelhardt, Antonio Sarikas.

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
