## [Decision Letter · Decision Letter 0]

14 Sep 2020

PONE-D-20-24341

Genetic ablation of Cullin-RING E3 ubiquitin ligase 7 restrains pressure overload-induced myocardial fibrosis

PLOS ONE

Dear Dr. Sarikas,

Thank you for submitting your manuscript to PLOS ONE. After careful consideration, we feel that it has merit but does not fully meet PLOS ONE’s publication criteria as it currently stands. Therefore, we invite you to submit a revised version of the manuscript that addresses the points raised during the review process.

All issues raised by expert reviewers are required.

We look forward to receiving your revised manuscript.

Kind regards,

Vincenzo Lionetti, M.D., PhD

Academic Editor

PLOS ONE

Journal Requirements:

2. Thank you for including your ethics statement:  "All animal studies were performed in accordance with the relevant guidelines and regulations of the responsible authorities (approval reference number 55.2-1-54-2532-160-13).".   

Please amend your current ethics statement to include the full name of the ethics committee that approved your specific study.

For additional information about PLOS ONE submissions requirements for ethics oversight of animal work, please refer to http://journals.plos.org/plosone/s/submission-guidelines#loc-animal-research  

Reviewers' comments:

Reviewer's Responses to Questions

**Comments to the Author**

1. Is the manuscript technically sound, and do the data support the conclusions?

Reviewer #1: Yes

Reviewer #2: Yes

2. Has the statistical analysis been performed appropriately and rigorously? 

Reviewer #1: Yes

Reviewer #2: Yes

3. Have the authors made all data underlying the findings in their manuscript fully available?

Reviewer #1: Yes

Reviewer #2: Yes

4. Is the manuscript presented in an intelligible fashion and written in standard English?

Reviewer #1: Yes

Reviewer #2: Yes

5. Review Comments to the Author

Reviewer #1: The paper of Anger et al presents a series of experiments aimed at characterizing the role of CRL7 in cardiac remodeling and fibrosis.

In particular, CM-specific knockout of Cul7 in two different mouse models results in significant reduction of interstitial fibrosis upon TAC, increased activation of PI3K/AKT axis, marked reduction of cellular apoptosis and significant downregulation of mRNA of Tgfb1.

The results obtained from this study visibly suggest that CRL7 activation promotes fibrotic remodeling of the myocardium following pressure overload stress, and indicate Cul7 as a possible therapeutic target to counteract this process.

The manuscript is well written and all the experiments are consistent with appropriate controls. Overall the results are assessed in a critical manner with some of the findings surprising but nonetheless interesting. Particularly intriguing is the evidence of the lack of functional cardiac improvement observed in Cul7-deficient mice, despite the remarkable reduction of fibrosis.

Although a deeper understanding of the mechanisms leading to this discrepancy is beyond the scope of this paper, I think that at least an analysis of biochemical and morphological markers of hypertrophy (fetal gene re-expression, myocytes cross sectional area) would be appropriate.

Additionally, the authors previously demonstrated that Cul7 downregulation improve the myocardium remodeling reactivates cardiomyocyte cell cycle and reduces hypertrophic cardiomyocyte growth in a mice model of permanent LAD ligation. In the present manuscript the authors highlight the antiapototic action of Cul-7 ablation but did not mention possible cell-cycle activation mechanisms. Although the pathological models are quite different, these discrepancies should be discussed.

Reviewer #2: This is a well done and interesting manuscript in which the authors conducted a series of experiments aimed at determining the role of Cullin-RING E3 ubiquitin 1 ligase 7 in cardiac hypertrophy. They have generated models of inducible genetic deletion in mice and subjected the animals to pressure overload induced hypertrophy to determine the effects on cardiac function under stress. Results show that this gene controls fibrosis as its absence ameliorate cardiac function, decreasing the expression of collagen by many folds as well as cardiac fibrosis and apoptosis.

I have no major questions, only some clarifications which could be added to the discussion.

1. How these results could be exploited for therapeutics is not clear. The authors may speculate on this.

2. Is there a link between production of collagen and apoptosis? Can the authors speculate also on the effects of AKT activation, which most probably plays an anti-apoptotic effect and its potentially negative effect on autophagy?

3. Since the KO has been performed in the cardiomyocyte compartment, is there a cross talk between the cardiomyocyte and the fibroblasts, which should be the primary source of collagen in the heart?

4. There are in the literature manuscripts demonstrating that AKT activation favors the capacity of the heart to cope with pressure overload stress. This effect is mediated also by an increased inotropic and, at the level of the myocyte, lusotropic effect. Have the authors tried to analyze cardiac function on isolated cardiomyocytes? Could they speculate on that?

5. AKT activation seems to decrease the level of inflammation which accompanies pressure-overload HF; due to the link between inflammation and fibrosis, this should be mentioned in the discussion.

6. PLOS authors have the option to publish the peer review history of their article (what does this mean?). If published, this will include your full peer review and any attached files.

Reviewer #1: No

Reviewer #2: No

---

## [Author Response · Author response to Decision Letter 0]

20 Nov 2020

Please see attached response letter to reviewer including additional figures and table.

---

## [Decision Letter · Decision Letter 1]

3 Dec 2020

Genetic ablation of Cullin-RING E3 ubiquitin ligase 7 restrains pressure overload-induced myocardial fibrosis

PONE-D-20-24341R1

Dear Dr. Sarikas,

We’re pleased to inform you that your manuscript has been judged scientifically suitable for publication and will be formally accepted for publication once it meets all outstanding technical requirements.

Kind regards,

Vincenzo Lionetti, M.D., PhD

Academic Editor

PLOS ONE

Additional Editor Comments (optional):

Reviewers' comments:

Reviewer's Responses to Questions

**Comments to the Author**

1. If the authors have adequately addressed your comments raised in a previous round of review and you feel that this manuscript is now acceptable for publication, you may indicate that here to bypass the “Comments to the Author” section, enter your conflict of interest statement in the “Confidential to Editor” section, and submit your "Accept" recommendation.

Reviewer #1: All comments have been addressed

Reviewer #2: All comments have been addressed

2. Is the manuscript technically sound, and do the data support the conclusions?

Reviewer #1: Yes

Reviewer #2: Yes

3. Has the statistical analysis been performed appropriately and rigorously? 

Reviewer #1: Yes

Reviewer #2: Yes

4. Have the authors made all data underlying the findings in their manuscript fully available?

Reviewer #1: Yes

Reviewer #2: Yes

5. Is the manuscript presented in an intelligible fashion and written in standard English?

Reviewer #1: Yes

Reviewer #2: Yes

6. Review Comments to the Author

Reviewer #1: (No Response)

Reviewer #2: I have no forte questions since all criticisms have been addressed in a coincise but extensive manner

7. PLOS authors have the option to publish the peer review history of their article (what does this mean?). If published, this will include your full peer review and any attached files.

Reviewer #1: No

Reviewer #2: No

---

## [Editor Report · Acceptance letter]

7 Dec 2020

PONE-D-20-24341R1 

Genetic ablation of Cullin-RING E3 ubiquitin ligase 7 restrains pressure overload-induced myocardial fibrosis 

Dear Dr. Sarikas:

I'm pleased to inform you that your manuscript has been deemed suitable for publication in PLOS ONE. Congratulations! Your manuscript is now with our production department. 

Kind regards, 

on behalf of

Prof. Vincenzo Lionetti 

Academic Editor

PLOS ONE